# Urinary β_2_-Microglobulin Predicts the Risk of Hypertension in Populations Chronically Exposed to Environmental Cadmium

**DOI:** 10.3390/jox15020049

**Published:** 2025-03-28

**Authors:** Soisungwan Satarug

**Affiliations:** Centre for Kidney Disease Research, Translational Research Institute, Woolloongabba, Brisbane, QLD 4102, Australia; sj.satarug@yahoo.com.au

**Keywords:** blood pressure regulation, β_2_-microglobulin, cadmium, eGFR, hypertension, kidneys

## Abstract

Chronic exposure to the pollutant cadmium (Cd) is inevitable for most people because it is present in nearly all food types. Concerningly, the risk of developing hypertension has been linked to dietary Cd exposure lower than 58 µg/day for a 70 kg person. The mechanisms involved are, however, unclear. Since the kidneys play an indispensable role in long-term blood pressure regulation, and they are also the main site of Cd accumulation and toxicity, a retrospective analysis was conducted to examine if kidney damage and malfunction, reflected by urinary β_2_-microglobulin excretion (E_β2M_), and the estimated glomerular filtration rate (eGFR), are related to Cd excretion (E_Cd_) and blood pressure variation. Data were obtained from 689 Thai Nationals without diabetes or occupational exposure to Cd, of which 32.4% had hypertension and 7.3% had β_2_-microglobulinuria, defined as an increase in the β_2_M excretion rate ≥ 300 µg/g creatinine. Respective prevalence odds ratio (POR) and 95% confidence interval (CI) values for β_2_-microglobulinuria and hypertension were 10.7 (1.36–83.4), *p* = 0.024 and 2.79 (1.60–4.87) *p* < 0.001, comparing the top quartile of E_Cd_ with the bottom quartile. Only in subjects with eGFR below 90 mL/min/1.73 m^2^ did systolic blood pressure (SBP) and diastolic blood pressure (DBP) both increase linearly with E_β2M_ (respective β = 0.182 and 0.192 for SBP and DBP) after adjustment for age, body mass index, gender, and smoking. The present study confirms the significant impact of Cd on the risk of having hypertension, following GFR loss induced by Cd. A simple mediation model analysis for cause–effect inference has provided, for the first time, evidence that may link rising SBP and DBP in Cd-exposed people to a novel role of β_2_M as a predictor of blood pressure variability.

## 1. Introduction

Nearly a third of the adult population worldwide is affected by hypertension, defined as a persistent elevation of systolic blood pressure (SBP) and/or diastolic blood pressure (DBP)  ≥ 140/90 mm Hg [1,2]. In 2019, high SBP contributed to 19.2% of all deaths globally [2]. Notably, hypertension is a modifiable risk factor for dementia and cardiovascular disease and is a contributing factor to the progression of chronic kidney disease (CKD) toward kidney failure [3,4,5]. The majority of hypertensive patients (90–95%) have primary hypertension, also known as essential hypertension, for which the cause is unknown, while an underlying pathology, which causes a rise in blood pressure, is responsible for hypertension development in 5–10% of hypertensive patients [3,4,5].

Cadmium (Cd) is one of the environmental contaminants with significant global public health threats because Cd is found in a normal diet [6,7,8]. Polluted air and cigarette smoke are increasingly reported as additional Cd exposure routes in urban populations [9,10,11]. Cd has no nutritional value or physiological role, but due to a lack of excretory mechanisms, it is retained within cells, tissues, and organs, especially the kidney cortex, where most acquired Cd accumulate [6,12,13].

Increased risks of kidney damage and dysfunction [14,15,16], hypertension [17,18,19], and CKD [20,21] in many populations have been associated with exposure to Cd levels much lower than a tolerable intake level of 58 µg/day for a 70 kg person, suggested to be an exposure level that carries a negligible health risk [https://apps.who.int/iris/handle/10665/44521, accessed on 7 February 2025]. However, there is evidence that links an increased risk of CKD with exposure to a Cd dose between 10 and 15 µg/day [20,21,22]. In a cross-section study on the U.S. general population, aged 20–85 years (*n* = 55,677), CKD risk rose linearly with blood Cd concentration independently of smoking and obesity [21]. In a prospective cohort study from the U.S., Cd excretion rates of ≥0.60 μg/g of creatinine or blood Cd levels of ≥0.70 μg/L were associated with an increased mortality in CKD patients [23].

Epidemiological studies from the U.S. [24,25,26], Canada [27], China [28,29], Korea [30,31], Japan [32], and Thailand [33,34] have implicated chronic exposure to Cd as an environmental risk factor hypertension. However, investigations into the mechanisms of Cd-linked hypertension are limited.

With the molecular weight of 12,000 Daltons, β_2_-microglobulin (β_2_M) passes through the glomerular membrane to lumen, where it is internalized and degraded by proximal tubular cells [35,36]. In Cd toxicity research, an increased excretion of β_2_M ≥ 300 µg/g creatinine is used to signify a reduced protein reabsorption [37]. Current evidence, however, suggest that β_2_M has a wide diversity of biological roles, which include blood pressure control [38,39,40]; an elevated plasma β_2_M has been found to be associated with increases in prevalent and incident hypertension in the Framingham Heart Study (*n* = 7065) [40].

The present study was aimed to explore the potential role of β_2_M in the variation in SBP/DBP and the risk of having hypertension in relation to eGFR and Cd body burden. In addition, it was aimed to examine if gender differences are related to Cd-induced hypertension.

## 2. Materials and Methods

### 2.1. Data Sources

Data were collected from 689 Thai Nationals (251 males, 438 females) who did not have diabetes type 2 [22,41,42]. Of a total 689 subjects, 308 were residents of Mae Sot District, Tak Province, endemically contaminated with Cd, and 381 persons were selected from the following three low-exposure localities: a non-contaminated zone of the Mae Sot District (*n* = 63), Bangkok (*n* = 248), and Nakhon-Si-Thammarat Province (*n* = 70). For the Bangkok group, those aged 16 years or older were selected [22]. The health status was ascertained by a physician’s examination reports and routine blood and urinary chemistry profiles. For the Nakhon-Si-Thammarat group [42] and the two Mae Sot groups, those who had resided at their current addresses for 30 years or longer were selected [22,41].

Exclusion criteria for all groups were pregnancy, breast-feeding, a history of exposure to metals in workplace settings, and a hospital record or physician’s diagnosis of an advanced chronic disease. The sociodemographic data, educational attainment, occupation, health status, family history of diabetes, and smoking status were obtained by structured interview questionnaires.

### 2.2. Measurement of Cd Exposure and Its Effects on Kidney Function

Urinary Cd and β_2_M were used to assess exposure to Cd and its effect on tubular re-absorptive function, respectively [22,41,42]. The estimated glomerular filtration rate (eGFR), a clinical measure of kidney function, was computed with equations of the Chronic Kidney Disease Epidemiology Collaboration (CKD-EPI) [43,44,45]. CKD stages 1, 2, 3, 4, and 5 corresponded to eGFR of 90–119, 60–89, 30–59, 15–29, and <15 mL/min/1.73 m^2^, respectively [45].

Samples of urine and whole blood with ethylene diamine tetra acetic acid (EDTA) as an anticoagulant were collected after an overnight fast. To prevent the degradation of β_2_M in acidic conditions, an alkaline (NaOH) solution was added to adjust the pH of urine samples to >6 before their storage. Aliquots of whole blood, plasma, and urine were stored at −80 °C for later analysis.

The alkaline picrate Jaffe’s reaction, as described by Bonness and Taussky [46] and Apple et al. [47], was used to determine urinary and plasma creatinine concentrations. The latex immunoagglutination method, as described by Bernard et al. [48], was employed to assay urinary β_2_M concentration (LX test, Eiken 2MGII, Eiken and Shionogi Co., Tokyo, Japan).

Urinary Cd concentration, expressed as µg/L was determined by Graphite Furnace Atomic Absorption Spectrometry (GBC System 5000, PAL2000 autosampler) [GBC Scientific Equipment, Hampshire, IL, USA], as described by Trzcinka-Ochocka et al. [49]. Multielement standards containing nine elements in dilute nitric acid, including As, Be, Cd, Cr (VI), Hg, Ni, Pb, Se, and Tl at 100 mg/L concentration per each element (Merck KGaA, Darmstadt, Germany) were used to calibrate the instrument. For quality control, the analytical accuracy and precision of Cd analysis were evaluated using reference urine metal controls containing Cd in three concentration ranges (Lyphocheck, Bio-Rad, Hercules, CA, USA). To maintain analytical accuracy, external quality assessments for Cd detection were conducted every three years.

For a urine sample containing Cd below the limit of detection (LOD), the Cd concentration assigned was the LOD value divided by the square root of 2 [50]. The LOD value for urinary Cd was the SD of at least 10 blank sample measurement multiplied by 3.

### 2.3. Normalization of Cd Excretion Rate and β_2_-Microglobulin Excretion Rate

Excretion of x (E_x_) was normalized to E_cr_ as [x]_u_/[cr]u, where x = Cd or β_2_M; [x]_u_ = urine concentration of x (mass/volume) and [cr]_u_ = urine creatinine concentration (mg/dL). E_x_/E_cr_ was expressed as an amount of x excreted per g of creatinine.

Excretion of x (E_x_) was normalized to creatinine clearance (C_cr_) as E_x_/C_cr_ = [Cd]_u_[cr]_p_/[cr]_u_, where x = Cd or β_2_M; [x]_u_ = urine concentration of x (mass/volume); [cr]_p_ = plasma creatinine concentration (mg/dL); and [cr]_u_ = urine creatinine concentration (mg/dL). E_x_/C_cr_ was expressed as an amount of x excreted per volume of the glomerular filtrate [51].

Normalizations of E_Cd_ and E_β2M_ were necessary because urine samples were collected at a single time point (term voided urine). The purpose of normalization of E_Cd_ and E_β2M_ to E_cr_ as E_Cd_/E_cr_ and E_β2M_/E_cr_ was to correct for differences in urine dilution among people. However, this E_cr_-normalization has been shown to introduce non-differential errors, which bias the dose–response relationship toward null [52,53]. In comparison, the purpose of normalization of E_Cd_ and E_β2M_ to C_cr_ as E_Cd_/C_cr_ and E_β2M_/C_cr_ was to simultaneously correct for interindividual differences in urine dilution, and the functioning nephrons. This C_cr_-normalization has unveiled an unambiguous effect of Cd on GFR.

### 2.4. Analysis of Mediated Effects

A simple mediation model is used to examine cause–effect relationships among study variables [54,55,56]. Here, it was used to determine if E_Cd_ could be causally related to rising SBP and DBP and if urinary β_2_M mediated Cd-induced SBP and DBP increment. Figure 1 depicts a generalized simple mediation model.

In the diagram above, A, B, and C’ are standardized β coefficients, describing an effect of X on Y with M as a mediator. These β values are used for computing, (1) total effect of X on Y designated as C, (2) the ratio of indirect effect, and (3) the proportion mediated [54,55,56].C = C’ + A*B

The ratio of indirect effect = A*B/C’

The proportion mediated = A*B/C’ + A*B

### 2.5. Statistical Analysis

Data were analyzed with IBM SPSS Statistics 21 (IBM Inc., New York, NY, USA). The Mann–Whitney U test. was used to assess the differences between males and females. The Kruskal–Wallis test was to assess differences across the quartiles of Eβ_2_M. The Pearson’s chi-squared test was used to determine differences in percentages and prevalences of smoking, hypertension, low eGFR, and β_2_-microglobulinuria. The conformity to a normal distribution of continuous variables was assessed by the one-sample Kolmogorov–Smirnov test. Logarithmic transformation was applied E_Cd_ and E_β2M,_ which showed a right-skewed distribution. The variable like eGFR, showed a left-skewed distribution and they were analyzed without any transformation. Determinants of SBP and DBP were identified by multiple linear regression modeling.

Determinants of the prevalence odds ratio (POR) values for hypertension and β_2_-microglobulinuria were obtained by logistic regression models, for which potential confounders (age, body mass index, smoking, and gender) were adjusted. For the E_cr_-normalized data, β_2_-microglobulinuria was defined as E_β2M_/E_cr_ ≥ 300 µg/g creatinine. For the C_cr_-normalized data, β_2_-microglobulinuria was defined as E_β2M_/C_cr_ ≥ 3 µg/L filtrate. For all tests, *p*-values ≤ 0.05 were considered to indicate statistical significance.

## 3. Results

### 3.1. Characterization of Selected Subjects According to eGFR and β_2_-Microglobulin Excretion

The description of participants can be found in Table 1.

Of a total 689 participants, the respective percentages (%) of females, smokers, and those with hypertension were 63.4, 28.3, and 32.4. The overall % of low eGFR and β_2_-microglobulinuria, defined as E_β2M_/E_cr_ ≥ 300 µg/g creatinine, and E_β2M_/C_cr_ ≥ 3 µg/L filtrate were 2.6, 7.3, and 6.8, respectively.

Respective overall mean values for age, BMI, eGFR, SBP, and DBP were 42.7 years, 23.4 kg/m^2^, 97.4, mL/min/1.73 m^2^, 121 mm Hg, and 78 mm Hg. The overall mean E_Cd_/E_cr_ and mean E_Cd_/C_cr_ were 2.77 µg/g creatinine and 0.0219 µg/L filtrate, respectively.

Given that E_β2M_ and eGFR have been shown to be inversely related, participants were assigned to the normal eGFR group or the reduced eGFR group. The normal eGFR group included those with eGFR values ≥ 90 mL/min/1.73 m^2^, while the reduced eGFR group enlisted those with GFR values < 90 mL/min/1.73 m^2^.

The parameters showing male–female differences in both normal and reduced eGFR groups were % smokers (particularly high among males), age, BMI, and Cd exposure levels (assessed as E_Cd_/E_cr_, and E_Cd_/C_cr_). In comparison, % hypertension was higher in females (30.7) than males (22.4) in the normal eGFR group only. Similarly, % β_2_-microglobulinuria, based on both E_cr_ and C_cr_ normalized data, were more prevalent in males compared to females in the normal eGFR group only. In the reduced eGFR group, however, the % β_2_-microgloubinuria and hypertension in males were similarly high as in females.

The inverse relationship between eGFR and E_β2M,_ were apparent also when participants were grouped according to the quartiles of β_2_-microglobulin excretion rate, assessed as E_β2M_/C_cr_ (Table 2).

The % of females, smokers, hypertension, and those with BMI values > 30 kg/m^2^ were similar across the E_β2M_/C_cr_ quartile groups. Mean BMI, mean SBP, and mean DBP were also similar across the E_β2M_/C_cr_ quartile groups. However, there were significant differences in % low eGFR, mean values for age, eGFR, E_Cd_/E_cr_, and E_Cd_/C_cr_ across E_β2M_/C_cr_ quartile groups. Notably, the prevalence of low eGFR in the E_β2M_/C_cr_ quartiles were markedly different (*p* < 0.001). In the top E_β2M_/C_cr_ quartile, the prevalence of low eGFR was as high as 10.3%, and it was nil in the bottom quartile. The prevalence of low eGFR in the E_β2M_/C_cr_ quartiles 2 and 3 were 0.7% and 1.4%, respectively.

### 3.2. Determinants of Risks of Hypertension and β_2_-Microglobulinuria

Two logistic regression models were constructed to evaluate an effect of Cd exposure on the POR values for two adverse outcomes (Table 3).

The POR for hypertension was influenced by age, BMI, and E_Cd_/C_cr_, but not gender or smoking. Per every one year older, the POR for hypertension increased 5.9% (95% CI: 4.0, 7.8, *p* < 0.001). The respective POR values for hypertension rose 2.08-fold (95% CI: 1.442, 3.001, *p* < 0.001) and 3.34-fold (95% CI: 1.481, 7.543, *p* = 0.004) in those with a BMI of 24–30 and > 30 kg/m_2_, compared to a BMI < 24 kg/m^2^. Compared with the bottom quartile, respective POR values for hypertension rose 2.52-fold (95% CI: 1.465, 4.342, *p* = 0.001) and 2.79-fold (95% CI: 1.595, 4.871, *p* < 0.001) in the E_Cd_/C_cr_ quartiles 3 and 2.

Among five independent variables tested, the POR for β_2_-microglobulinuria was affected markedly by Cd (E_Cd_/C_cr_). Compared with the bottom quartile of E_Cd_/C_cr_, the POR for β_2_-microglobulinuria in the top quartile rose 10.67-fold (95% CI: 1.364, 83.4, *p* = 0.024). However, the 6.98-fold (95% CI: 0.887, 54.93) increase in POR for β_2_-microglobulinuria in the E_Cd_/C_cr_ quartile 3 was statistically significant (*p* = 0.065).

Scatterplots were constructed to visualize the correlations of SBP and DBP with a β_2_-microglobulin excretion rate only in the female gender (Figure 1).

Additional scatterplots (Figure 2) indicate the correlations of SBP and DBP with a β_2_-microglobulin excretion rate only in the reduced eGFR group.

### 3.3. Urinary β_2_M Predicts Blood Pressure Elevation

In multiple linear regressions of SBP and DBP (Table 4), respective proportions of the SBP variation in all subjects, the normal eGFR group, and the reduced eGFR group that were related to a set of six independent variables (age, BMI, E_Cd_/Ccr, E_β2M_/C_cr_, smoking, and gender) were 14.5% (*p* < 0.001), 10.6% (*p* < 0.001), and 10.3% (*p* < 0.001). In comparison, the same set of independent variables explained only 5.4% (*p* < 0.001) and 5.3% (*p* < 0.001) of the DBP variation in all subjects and the normal eGFR group. The DBP variability among those with a reduced eGFR was minimally related to these variables.

Notably, however, the associations of E_β2M_/C_cr_ with SBP (β = 0.182, *p* = 0.013) and DBP (β = 0.192, *p* = 0.012) were found only in the reduced eGFR group. These results were in line with scatterplots in Figure 2.

### 3.4. GFR and β_2_-Microglobulin as Mediators of Cd-Induced Blood Pressure Elevation

Figure 3 provides results of a simple mediation model that assessed whether eGFR mediated an effect of Cd on SBP and DBP.

In the SBP model (Figure 3), Cd had a direct effect on a mediator (eGFR) (β =−0.116, *p* < 0.001) as well a significant indirect effect on SBP (Sobel test *p* = 0.006) (Figure 3a,c). A fall of eGFR mediated the indirect effect of Cd on SBP. The proportion mediated by a declining eGFR, calculated from the equation (A*B/C’ + A*B), was 19%.

In the DBP model, Cd had a direct effect a mediator (eGFR) (β =−0.133, *p* < 0.001) as well an indirect effect (Sobel test *p* = 0.012) (Figure 3b,d). The indirect effect of Cd on DBP was also through a reduction in eGFR. Using the equation (A*B/C’ + A*B), the proportion of effect mediated through eGFR was 16%.

Mediating effects of Cd on SBP and DBP by β_2_M were assessed in Figure 4a and 4b, respectively. In the SBP model, Cd had a direct effect on a mediator (β_2_M) (β = 0.294, *p* < 0.001) (Figure 4a). The indirect effect of Cd on SBP was statistically significant (Sobel test *p* = 0.007) (Figure 4c). In the DBP model, Cd had a direct effect on a mediator (β_2_M) (β = 0.296, *p* < 0.001), as well as a significant indirect effect on SBP (Sobel test *p* = 0.024) (Figure 4b,d). The effects of Cd on SBP and DBP were mediated through β_2_M.

## 4. Discussion

This study analyzed data from 251 males and 438 females, aged 16–80 years, who did not have diabetes type 2. They were selected from a large cohort of apparently healthy Thais. The % females, smokers, obese subjects, hypertension, β_2_-microglubulinuria, and CKD (eGFR ≤ 60 mL/min/1.73 m^2^) were 63.4, 28.3, 4.9, 32.4, 7.3, and 2.6, respectively. Mean BMI, mean eGFR were 23.4 kg/m^2^, and 97.4 mL/min/1.73 m^2^, respectively. Age and BMI histograms of this study group conformed to a normal distribution. This group could thus be considered to represent a sample of the general population. The mean E_Cd_/E_cr_ and mean E_Cd_/C_cr_ were 2.77 µg/g creatinine and 0.0219 µg/L filtrate, respectively.

The environmental Cd exposure levels among cohort participants were comparable to those reported for non-occupationally exposed populations. It is well recognized that blood pressure levels and prevalence of hypertension in males and females are dissimilar [3,4,57,58,59]. The present study included 438 females and 251 males. A small number of males was a limitation because it precluded a definitive conclusion on an effect of Cd in men. A one-time-only assessment of Cd exposure and its effects was also its limitation.

### 4.1. Cd and Risks of Developing Hypertension and β_2_-Microglobulinuria

In line with previous studies [24,25,26,27], age, BMI, and Cd exposure were associated with an increased risk of having hypertension (Table 3). The risk of having hypertension rose 5.9% for every one-year older, and it was increased 3.3-fold in those with BMI > 30 kg/m^2^. There was a 2.5-fold increase in the risk of hypertension at (E_Cd_/C_cr_) × 100 value of 0.71 µg/L filtrate, corresponding to 0.68 µg/g creatinine. In comparison, however, neither age nor BMI influenced the risk of having tubular proteinuria. An increase in risk of having such proteinuria was statistically significant only in the top quartile of Cd exposure (E_Cd_/C_cr_) × 100 value of 2.74 µg/L filtrate, corresponding to 2.32 µg/g creatinine. These data suggest that β_2_-microglubulinuria occurred after hypertension development; a significant increase in risk of hypertension was observed at the exposure level producing E_Cd_/E_cr_ of 0.68 µg/g creatinine, which was much lower than E_Cd_/E_cr_ of 2.32 µg/g creatinine. Thus, blood pressure increase could be a sensitive biomarker for an early detection of an adverse effect of Cd on kidneys.

### 4.2. Effects of Cd on SBP/DBP and Tubular Function

In the SBP regression model, including all subjects (Table 4), SBP did not correlate with E_Cd_ or E_β2M_, but it was associated with age (β = 0.323), BMI (β = 0.188), and gender (β = 0.105). The DBP regression model showed similar correlative patterns, but the strengths of correlation were weaker, compared with SBP. In subgroup analysis, association of SBP with E_β2M_ (β = 0.182) and DBP with E_β2M_ (β = 0.192) were found only in the reduced eGFR group. These results were evident also from scatterplots in Figure 2. Similarly, a 10% increase in β_2_M excretion rate was found in participants of the SPRINT Trial (*n* = 2436) who had low eGFR and a 16 mm Hg higher SBP than mean SBP [60].

A mediated effect analysis (Figure 3) has revealed that respective SBP and DBP increases by 19% and 16% were mediated by a declining eGFR. This explained the presence of SBP/E_β2M_ and DBP/E_β2M_ correlations only in those with eGFR < 90 mL/min/1.73 m^2^. Previously, GFR loss was found to mediate the elevation of SBP and DBP in people exposed to relatively higher Cd doses than participants in the present study [34].

Interestingly, the effects of Cd exposure on SBP and DBP among participants in the present study appeared to be mediated by β_2_M (Figure 4). This mediating effect of β_2_M provided an explanation for the absence of SBP-E_Cd_ and DBP-E_Cd_ correlations in multiple regression analysis (Table 4). Further studies are warranted to examine the played by β_2_M in Cd-induced hypertension.

In a cross-sectional study on Japanese general population, an increased risk of hypertension was associated an elevated β_2_M excretion [61]. In a prospective cohort study on the Japanese population, a 79% increase in the risk of eGFR reduction of 10 mL/min/1.73 m^2^ within five years was found in cohort participants who had β_2_-microglobulinuria [62].

### 4.3. Gender Differences in Blood Pressure Variability and Cd Effects

In the descriptive characteristics of cohort participants (Table 1), % hypertension in the normal eGFR group was higher in females than males (30.7 vs. 22.4). In the reduced eGFR group, however, the % hypertension and % β_2_-microgloubinuria in males and females were similar.

Further evidence for gender differences in risk of hypertension was apparent from a multiple regression analysis (Table 4) where SBP, not DBP, was found to be associated with gender (β = 0.105) after adjustment for effects of other variables (age, BMI, smoking). The scatterplots in Figure 1 revealed significant SBP-E_β2M_ and DBP-E_β2M_ correlations in females.

The different effects of Cd on blood pressure in men and women have been noted in several studies. For example, 1.54-time and 2.38-time increases in hypertension risk were found in white and Mexican-American women, who had blood Cd as little as 0.4 µg/L, compared with black women, white, black, or Mexican-American men [26]. In a study from Taiwan, Cd-related kidney damage, evident from an elevated excretion rate of N-acetyl-β-D-glucosaminidase (NAG), was found in women only [63].

Previously, gender differentiated effects of Cd on blood pressure were attributable to circulating levels of sex hormones [64,65,66,67]. A rise in urinary Cd levels from <2 to ≥3 μg/g creatinine in postmenopausal Japanese women was associated with an elevation of serum testosterone by 28% [64]. Serum estradiol levels in postmenopausal Japanese and Swedish women showed an inverse association with urinary Cd excretion rates [65,66]. The Swiss Kidney Project on Genes in Hypertension observed a correlation between urinary Cd and testosterone excretion in men, while a trend for an association was found in women [67].

In general, hypertension occurs more frequently in men than premenopausal women of a similar age [3,4,57,58,59]. Clinical and experimental studies have attributed male preponderance hypertension to the eicosanoid, 20-hydroxyeicosatetraenoic acid (20-HETE), which mediates an effect of the androgen hormone on SBP and DBP levels [68,69,70,71]. Studies from Thailand support the role of 20-HETE in Cd-induced hypertension; an association of urinary 20-HETE and urinary Cd was found in women only, while urinary 20-HETE and SBP were associated with an increased albumin excretion in men [33].

### 4.4. Modifiable Risk by Other Metals

In addition to the influence of the sex hormone, potential roles of other metals, like zinc (Zn) and calcium (Ca), have been reported in humans [72] and experimental animals [73,74]. Given their similar electronegativity, Zn, in theory, can modulate the manifestation of nephrotoxicity of Cd. In a study from the U.S. (*n* = 9557), aged ≥ 18 years, there was a 2.79-fold increase in the risk of having CKD among those with blood Cd in quartile 4, compared to those with blood Cd in the quartile 1 [72]. Comparing low zinc and high Cd with high zinc plus high Cd and high zinc plus low Cd, there were 90% and 89% reductions in CKD risk, respectively.

An elevated systemic blood pressure and a decline in kidney function, measured by inulin clearance, were observed in rats fed with a diet containing Zn that was 40 times higher than that of a normal diet for 4 weeks [73]. Ca and Cd have similar ionic radii, and in theory, Ca can be expected to influence Cd-induced hypertension. Such an effect of Ca has been demonstrated experimentally [74].

## 5. Conclusions

Exposure to environmental Cd, producing a urinary Cd excretion rate of 0.68 µg/g creatinine, is associated with a 2.5-fold increase in risk of developing hypertension. The development of hypertension in Cd-exposed people is an outcome of rising SBP and DBP because of a decrease in eGFR induced by Cd. The increases in SBP and DBP by 16–19% are found to be causally related to a fall of eGFR. This eGFR loss occurs in advance of β_2_-microglobulinuria, given that an increased risk of having β_2_-microglobulinuria is observed at a urinary Cd excretion rate of 2.32 µg/g creatinine. There is evidence that β_2_M independently affects blood pressure.

## Data Availability

All data are contained within this article.

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
