# Peer review of "Urinary β2-Microglobulin Predicts the Risk of Hypertension in Populations Chronically Exposed to Environmental Cadmium"

_jox, 2025, doi:10.3390/jox15020049_

Round 1

Reviewer 1 Report

Comments and Suggestions for Authors

This manuscript reports on the detailed analysis of previously published results from 689 study subjects whose urine had been analyzed for Cd, b2microglobulin and creatinine. It is suggested to broaden the scope of the introduction and the discussion as only human environmental Cd exposure is attempted to be linked to hypertension without considering other potential contributing/confounding factors. It would improve the quality of the manuscript if some other factors (i.e. other environmental pollutants) are identified that may contribute to the human risk of hypertension especially since the latter has a complex etiology that involves both genetic and non-genetic factors, such a heat-stress, dehydration, exposure to agrochemicals, heavy metals and use of hard water, infections, mycotoxins, nephrotoxic agents, altitude (Pathophysiology 2024, 31 (4), pp.761-786). In addition, geogenic factors such as Ca intake have also been suggested to significantly effect the CD-mediated induction of hypertension in animal models (Proc. Natl. Acad. Sci. USA, 1981, 78 (10), 6494-6498). If this major deficiency as well as some other ones are addressed (see detailed comments below), the manuscript can be re-evaluated for publication.

Detailed comments

Abstract

Line 12: It should read ‘if kidney damage’.

Introduction

Line 37: Please specify if the increase in the ranking of CKD from 13th to 10th place pertains to a region (e.g. Asia) or worldwide.

Line 43: With regard to ‘albuminuria’ please specify the albumin conc. in urine that needs to be exceeded to increase clarity.

Line 45: It should read ‘eGFR falling’.

Line 52: The molecular weight of b2-microglobulin in 12,000. It is therefore suggested to replace ‘low-molecular-weight’ with ‘12 kDa’ protein.

Line 54-55: A better formulation would be ‘b2M serves a variety of biological roles, which includes…’. ‘SH2B3’ is entirely unclear. Please explain what this abbreviation means.

Line 61: After ‘hypertension’ it should be specified how this was defined (e.g. systolic blood pressure ≥140 mm Hg). After ‘b2-microglobinuria’ it should be added how this is defined (e.g. urinary b2M > xy ug/L).

Materials and Methods

Line 83: It should read ‘mm Hg’.

Line 92: It should read ‘for the concentration’.

Line 100” ‘urine metal control levels 1, 2 and 3’ is unclear. Please reformulate so it is clear what is meant here. It appears that this statement refers to the use of some sort of standard reference materials that were obtained from Lyphcheck.

Line 110: To increase clarity please provide the equations in brackets here (not below).

Line 111: It should read ‘normalizations’.

Line 149: ‘showed a right-skewed distribution’ is unclear and should be better explained (please provide context).

Line 150: ‘showed a left-skewed distribution’ is unclear and should be better explained (please provide context).

Table 1: Under parameters it seems that a number before ‘ml/min/1.73 m2’ is missing. After ‘Systolic blood pressure’ and ‘Diastolic blood pressure’ please add the abbreviations ‘SBP’ and DBP, respectively.

Table 2: It is unclear why the total number of participants is lower (n=572) than in Table 1 (n=689). Please provide an explanation in a footnote of Table 2.

Line 226: It should read ‘2.79-fold’.

Line 239: Please define the abbreviation ‘SOP’ and ‘DBP’.

Line 274: It should read ‘In the SBP model’.

Line 282: It should read ‘whether urinary Cd’.

Line 288: It should read ‘In the SBP model’.

Line 289: It should read ‘In the DBP model’.

Line 292: ‘were mediated completely’ is unclear. Please reformulate.

Line 305: ‘inequal number of males and females recruited’ needs to be reformulated as it does not make sense in the context of the sentence.

Line 336: It should read ‘effects of urinary Cd on SBP’.

Line 339: It should read ‘Further studies are warranted’.

Line 341: ‘SH2B3’ is entirely unclear. Please explain what this abbreviation means as this is the first time this abbreviation is used.

Line 345: It would help if the definition of how the authors of this study defined b2-microglobinuria.

Line 348: It should read ‘(30.7 vs 22.4)’.

Line 357: It should read ‘from the U.S. reported’.

Line 378: It should read ‘Thailand, which reported an’.

Line 286: It should read ‘This eGFR’.   

Author Response

Reviewer 1

Comments and Suggestions

This manuscript reports on the detailed analysis of previously published results from 689 study subjects whose urine had been analyzed for Cd, b2microglobulin and creatinine. It is suggested to broaden the scope of the introduction and the discussion as only human environmental Cd exposure is attempted to be linked to hypertension without considering other potential contributing/confounding factors. It would improve the quality of the manuscript if some other factors (i.e. other environmental pollutants) are identified that may contribute to the human risk of hypertension especially since the latter has a complex etiology that involves both genetic and non-genetic factors, such a heat-stress, dehydration, exposure to agrochemicals, heavy metals and use of hard water, infections, mycotoxins, nephrotoxic agents, altitude (Pathophysiology 2024, 31 (4), pp.761-786). In addition, geogenic factors such as Ca intake have also been suggested to significantly effect the CD-mediated induction of hypertension in animal models (Proc. Natl. Acad. Sci. USA, 1981, 78 (10), 6494-6498). If this major deficiency as well as some other ones are addressed (see detailed comments below), the manuscript can be re-evaluated for publication.

RESPONSE: Thank you for a thorough evaluation of my work, comments, and suggestions to improve a paper.  All issues the reviewer raised have been addressed and resultant changes to the text are in blue.  Point-by-point responses to comments are provided below.

Comment 1: It is suggested to broaden the scope of the introduction and the discussion as only human environmental Cd exposure is attempted to be linked to hypertension without considering other potential contributing/confounding factors (Pathophysiology 2024, 31 (4), pp.761-786).

RESPONSE: Thank you for given me an opportunity to clarify the objective of the present work. It is indisputable that many environmental, occupational, and life-style related factors contribute to the development of hypertension, as addressed in a review, Pathophysiology 2024, 31 (4), pp.761-786.  In the present study, however, the focus was on the mechanism by which Cd induces hypertension, using a simple mediation model analysis (Scheme 1).  Evidence that low environmental Cd exposure is a contributing factor in hypertension development, CKD onset and progression to kidney failure in the general population is compelling, but research studies to connect these outcomes are limited. Other environmental risk factors such as dehydration, heat stress, are outside the scope of this paper.

Comment 2: Geogenic factors such as Ca intake have also been suggested to significantly affect the Cd mediated induction of hypertension in animal models (Proc. Natl. Acad. Sci. USA, 1981, 78 (10), 6494-6498).

RESPONSE: Thank you for raising an important issue of influences from other metals, notably Ca, that may affect Cd-induced hypertension.  A new subsection, 4.3. Modifiable Risk by Other Metals, has been inserted in the Discussion (lines 381-395) and the reference, Proc. Natl. Acad. Sci. USA, 1981, 78 (10), 6494-6498) has now been cited as Ref. 74 plus three additional references [Ref. 71-73].

4.3. Modifiable Risk by Other Metals

In addition to the influence of sex hormone, potential roles of other metals, like zinc and calcium have been reported in humans [71,72] and experimental animals [73,74]. Given their similar electronegativity, zinc, in theory, can modulate the manifestation of nephrotoxicity of Cd. In a cohort of a sample group of 9557 participants, aged ≥ 18 years, equivalent to 236,263,413 community-dwelling U.S. adults, there was a 2.79-fold increase in the risk of having CKD among those in the top quartile of blood Cd, compared to the bottom quartile [71]. The risk of having CKD was reduced by 90% and 89%, comparing low zinc and high Cd with high zinc plus high Cd and high zinc plus low Cd, respectively.

Rising systemic blood pressure and declining kidney function, measured by inulin clearance have been observed in rats fed with a diet containing Zn that was 40 times higher than that of a normal diet for 4 weeks [73]. Calcium and Cd have similar ionic radii, and in theory, calcium can be expected to influence Cd-induced hypertension. Such effect of calcium has been demonstrated experimentally [74].

[71] Lin, C.J.; Shih, H.M.; Wu, P.C.; Pan, C.F.; Lin, Y.H.; Wu, C.J. Plasma selenium and zinc alter associations between ne-phrotoxic metals and chronic kidney disease: Results from NHANES database 2011–2018. Ann. Acad. Med. Singap. 2023, 52, 398-410.

[72] Satarug, S. Antioxidative Function of Zinc and Its Protection Against the Onset and Progression of Kidney Disease Due to Cadmium. Biomolecules 2025, 15, 183.

[73] Yanagisawa, H.; Miyazaki, T.; Nodera, M.; Miyajima, Y.; Suzuki, T.; Kido, T.; Suka, M. Zinc-excess intake causes the deterioration of renal function accompanied by an elevation in systemic blood pressure primarily through superoxide radical-induced oxidative stress. Int. J. Toxicol. 2014, 33, 288-296.

[74] Revis NW, Zinsmeister AR, Bull R. Atherosclerosis and hypertension induction by lead and cadmium ions: an effect prevented by calcium ion. Proc. Natl. Acad. Sci. USA. 1981, 78, 6494-6498.

Detailed comments

Abstract

Line 12: It should read ‘if kidney damage’.

RESPONSE: A typo error has been corrected. Part of Abstract has been rewritten to provide a definition for β2-microglobulinuria and to emphasize a novel role of β2M as a mediator of Cd-induced hypertension. State it differently, Cd induces hypertension is through β2M.  A revised abstract is quoted below.

Abstract: Exposure to the environmental pollutant cadmium (Cd) is inevitable for most people because of its ubiquitous presence in nearly all food types. Concerningly, the risk of developing hypertension has been linked to dietary Cd exposure. The mechanisms involved are however, unclear. Because the kidneys play an indispensable role in long-term blood pressure regulation, and they are also the principal site of Cd accumulation and toxicity, a retrospective analysis was conducted to examine if kidney damage and malfunction, reflected by urinary β2-microglobulin excretion (Eβ2M), and the estimated glomerular filtration rate (eGFR), are related to Cd excretion (ECd) and blood pressure variation. Data were obtained from 689 Thai Nationals without diabetes or occupational exposure to Cd, of which 32.4% had hypertension, and 7.3 % had β2-microglobulinuria, defined as an increase of β2M excretion rate ≥ 300 µg/g creatinine. People in the top quartile of ECd had much higher risks of having β2-microglobulinuria [POR = 10.7 (95%CI:1.36, 83.4), p = 0.024), and hypertension [POR = 2.79 (95% CI:1.60, 4.87), p <0.001]. Only in subjects with eGFR below 90 mL/min/1.73 m2, systolic blood pressure (SBP) and diastolic blood pressure (DBP) both increased linearly with Eβ2M (respective β = 0.182 and 0.192 for SBP and DBP) after adjustment for age, body mass index, gender, and smoking. The present study confirms the significant impact of Cd on risk of having hypertension, following GFR loss induced by Cd. A simple mediation model analysis has provided, for the first time, evidence that may link rising SBP and DBP in Cd-exposed people to a novel role of β2M in blood pressure regulation.

Introduction

Line 37: Please specify if the increase in the ranking of CKD from 13th to 10th place pertains to a region (e.g. Asia) or worldwide.

Line 43: With regard to ‘albuminuria’ please specify the albumin conc. in urine that needs to be exceeded to increase clarity.

Line 45: It should read ‘eGFR falling’.

Line 52: The molecular weight of b2-microglobulin in 12,000. It is therefore suggested to replace ‘low-molecular-weight’ with ‘12 kDa’ protein.

Line 54-55: A better formulation would be ‘b2M serves a variety of biological roles, which includes…’. ‘SH2B3’ is entirely unclear. Please explain what this abbreviation means.

Line 61: After ‘hypertension’ it should be specified how this was defined (e.g. systolic blood pressure ≥140 mm Hg). After ‘b2-microglobinuria’ it should be added how this is defined (e.g. urinary b2M > xy ug/L).

RESPONSES:

  • Typo errors have been corrected. Necessary rewording has bene undertaken, where applicable to improve clarity.
  • Changes to the text are in blue.

Materials and Methods

Line 83: It should read ‘mm Hg’.

Line 92: It should read ‘for the concentration’.

Line 100” ‘urine metal control levels 1, 2 and 3’ is unclear. Please reformulate so it is clear what is meant here. It appears that this statement refers to the use of some sort of standard reference materials that were obtained from Lyphcheck.

Line 110: To increase clarity please provide the equations in brackets here (not below).

Line 111: It should read ‘normalizations’.

Line 149: ‘showed a right-skewed distribution’ is unclear and should be better explained (please provide context).

Line 150: ‘showed a left-skewed distribution’ is unclear and should be better explained (please provide context).

Table 1: Under parameters it seems that a number before ‘ml/min/1.73 m2’ is missing. After ‘Systolic blood pressure’ and ‘Diastolic blood pressure’ please add the abbreviations ‘SBP’ and DBP, respectively.

Table 2: It is unclear why the total number of participants is lower (n=572) than in Table 1 (n=689). Please provide an explanation in a footnote of Table 2.

Line 226: It should read ‘2.79-fold’.

Line 239: Please define the abbreviation ‘SOP’ and ‘DBP’.

Line 274: It should read ‘In the SBP model’.

Line 282: It should read ‘whether urinary Cd’.

Line 288: It should read ‘In the SBP model’.

Line 289: It should read ‘In the DBP model’.

Line 292: ‘were mediated completely’ is unclear. Please reformulate.

Line 305: ‘inequal number of males and females recruited’ needs to be reformulated as it does not make sense in the context of the sentence.

Line 336: It should read ‘effects of urinary Cd on SBP’.

Line 339: It should read ‘Further studies are warranted’.

Line 341: ‘SH2B3’ is entirely unclear. Please explain what this abbreviation means as this is the first time this abbreviation is used.

Line 345: It would help if the definition of how the authors of this study defined b2-microglobinuria.

Line 348: It should read ‘(30.7 vs 22.4)’.

Line 357: It should read ‘from the U.S. reported’.

Line 378: It should read ‘Thailand, which reported an’.

Line 286: It should read ‘This eGFR’.  

RESPONSES:

  • Typo errors have been corrected. Necessary rewording has bene undertaken, where applicable to improve clarity.
  • Changes to the text are in blue.

Reviewer 2 Report

Comments and Suggestions for Authors

This study uses a previously generated data base to analyze for associations between the toxin cadmium and beta2-microglobulin levels and extent of hypertension.  The analysis is straightforward, and the results are what would be expected.  However, the results should be of interest to a reasonable percentage of the readership.

Please attempt to get each table and figure with their caption on a single page (if possible).

Line 55, "role" should be the plural "roles".

Indicate if the informed consent was written (as it presumably was).

First paragraph of 4.1.  If these effects have been observed previously (as they undoubtedly have), this should be noted with appropriate citations.

Author Response

Reviewer 2

Comments and Suggestions

This study uses a previously generated data base to analyze for associations between the toxin cadmium and beta2-microglobulin levels and extent of hypertension.  The analysis is straightforward, and the results are what would be expected.  However, the results should be of interest to a reasonable percentage of the readership.

RESPONSE: Thank you for evaluating my work, comments, and suggestions to improve a paper.

Comment 1. Please attempt to get each table and figure with their caption on a single page (if possible).

RESPONSE:  Paragraphs have been rearranged, where possible, to accommodate each table and figure with their caption on a single page.

Comment 2. Line 55, "role" should be the plural "roles".

RESPONSE: A correction has been undertaken.

Comment 3. Indicate if the informed consent was written (as it presumably was).

RESPONSE: Written informed consent has been indicated.

Comment 4. First paragraph of 4.1.  If these effects have been observed previously (as they undoubtedly have), this should be noted with appropriate citations.

RESPONSE: The referred statements have been changed to read as below and pertinent references  have now been cited.

“In line with previous studies [23-31], age, BMI and Cd exposure were associated with an increased risk of having hypertension (Table 3). The risk of having hypertension rose 5.9% for every one-year older, and it was increased 3.3-fold in those with BMI > 30 kg/m2. There was a 2.5-fold increase in risk of hypertension at (ECd/Ccr) ×100 value of 0.71 µg/L filtrate, corresponding to 0.68 µg/g creatinine.”

Reviewer 3 Report

Comments and Suggestions for Authors

First of all, we would like to thank the authors for a very interesting manuscript on the use of urinary B2M as a predictor of hypertension risk in a population chronically exposed to cadmium.

Urinary B2M has emerged as a useful biomarker in the assessment of chronic cadmium (Cd) exposure and its health effects, especially with regard to the risk of high blood pressure.  B2M, a low molecular weight protein that leaches into the kidneys, is considered an indicator of proximal tubular kidney damage. In individuals chronically exposed to cadmium, elevated urinary B2M levels reflect a possible impairment of renal function.

Therefore, the hypothesis of the work as well as the objectives are correct and interesting. The manuscript is well written and drafted, concise and clear, however there are some nuances that should be improved for publication. 

1.- The paragraph between lines 36 and 40 should be removed, it does not add to the introduction to talk about something that is already known about CKD mortality; it would be more relevant to mention the data on CKD associated with chronic cadmium exposure and its relevance to mortality.

2.- The paragraph between lines 41 to 46 is more of the same, it is repetitive to mention the definition of CKD; it is well known, it does not add anything to the introduction. Please remove it.

3.- The text from lines 59 to 62 should be moved to the section on materials and methods.

4.-The participant selection section is a bit tedious, it would look better with a flow chart adding the exclusion and inclusion criteria. 

5.- The sections on results, discussion and conclusions seem clear to me, I have no further comments here.

Author Response

Reviewer 3

Comments and Suggestions

First of all, we would like to thank the authors for a very interesting manuscript on the use of urinary B2M as a predictor of hypertension risk in a population chronically exposed to cadmium.

Urinary B2M has emerged as a useful biomarker in the assessment of chronic cadmium (Cd) exposure and its health effects, especially with regard to the risk of high blood pressure.  B2M, a low molecular weight protein that leaches into the kidneys, is considered an indicator of proximal tubular kidney damage. In individuals chronically exposed to cadmium, elevated urinary B2M levels reflect a possible impairment of renal function.

Therefore, the hypothesis of the work as well as the objectives are correct and interesting. The manuscript is well written and drafted, concise and clear, however there are some nuances that should be improved for publication.

RESPONSE: Thank you for reviewing my work, knowledgeable comments, and suggestions to improve a paper.

Comment 1. The paragraph between lines 36 and 40 should be removed, it does not add to the introduction to talk about something that is already known about CKD mortality; it would be more relevant to mention the data on CKD associated with chronic cadmium exposure and its relevance to mortality.

RESPONSE: The referred paragraph has now been replaced with association of CKD with chronic cadmium exposure and its relevance to mortality that the reviewer has suggested (lines 37-44), quoted below.

“Indeed, evidence that links an enhanced risk of having CKD with low environmental Cd exposure is particularly compelling [15–18]. In a cross-section study on U.S. general population, aged 20-85 years (n = 55,677), the risk of having CKD increased 2.1-fold, 3.2-fold, and 5.5-fold, comparing blood Cd levels of rose 0.21–0.35, 0.36–0.60, and > 0.60 µg/L with blood Cd levels < 0.21 to respectively [16]. The reported increased CKD risks were adjusted for smoking effects [16]. In a prospective cohort study, the mortality among the U.S. adults with CKD rose with an elevated exposure to environmental Cd, reflected by Cd excretion rates of ≥ 0.60 μg/g of creatinine or blood Cd levels of ≥ 0.70 μg/L [19].”

Comment 2. The paragraph between lines 41 to 46 is more of the same, it is repetitive to mention the definition of CKD; it is well known, it does not add anything to the introduction. Please remove it.

RESPONSE: The referred statements have been replaced with below sentence (lines 45-47) with three additional references (ref. 20-22).

“Hypertension, defined as a systolic blood pressure (SBP) of ≥ 140 mmHg or a diastolic blood pressure (DBP) of ≥90 mmHg, is one of the most widely recognized consequences of kidney damage [20-22].”

[20] Crowley, S.D.; Coffman, T.M. The inextricable role of the kidney in hypertension. J. Clin. Investig. 2014, 124, 2341–2347.

[21] Ohashi, N.; Ishigaki, S.; Isobe, S. The pivotal role of melatonin in ameliorating chronic kidney disease by suppression of the renin-angiotensin system in the kidney. Hypertens. Res. 2019, 42, 761–768.

[22] Aoki, T.; Ohashi, N.; Isobe, S.; Ishigaki, S.; Matsuyama, T.; Sato, T.; Fujikura, T.; Kato, A.; Miyajima, H.; Yasuda, H. Chronotherapy with a Renin-angiotensin System Inhibitor Ameliorates Renal Damage by Suppressing Intrarenal Ren-in-angiotensin System Activation. Intern. Med. 2020, 59, 2237–2244.

Comment 3. The text from lines 59 to 62 should be moved to the section on materials and methods.

RESPONSE: The referred text has been moved to Section 2.1. Participant Selection (line 65-67).

Comment 4. The participant selection section is a bit tedious, it would look better with a flow chart adding the exclusion and inclusion criteria.

RESPONSE:  A flow-chart was not applicable because participants in the parent cohorts were recruited from different geographic regions of Thailand at different times with somewhat variable inclusion criteria. The selected participants formed a sample that could be considered as representative of non-diabetics Thai population.

Comment 5. The sections on results, discussion and conclusions seem clear to me, I have no further comments here.

RESPONSE: Thank you.

Round 2

Reviewer 1 Report

Comments and Suggestions for Authors

The author has adequately addressed my previous suggestions to improve the overall clarity of the manuscript. There are, however,  two remaining major deficiencies. The first is the failure to provide the reader with a reference or website which contains the data from the original study that were re-analyzed in the present study. While some cryptic information is provided as to how It is unclear to the reader what methods was analyzed to analyze blood and urine for b2-microglobulin. Secondly, any details that pertain to the analytical instruments that were employed to analyze the blood and urine samples are missing. These details must have been reported in the original study and should be briefly summarized in the present study, either in form of a Table and or electronic supplementary information. It is critical to provide the reader with detailed information about how the analytical data that are analyzed in the present study were generated as otherwise the analysis of the data could be inherently flawed. After addressing these major issues, the author should address some additional minor deficiencies before the manuscript can be accepted for publications.  

Detailed comments

Abstract

Line 9: It is suggested to provide the corresponding threshold Cd dose/day that links human Cd exposure with hypertension (e.g. ≥50 μg Cd per day).

Line 17+18: The abbreviation ‘POR’ as well as the 4th number after ‘POR’ must be explained.

Line 22: It would be helpful if the phrase ‘mediation model analysis’ could be briefly explained.

Line 23: It should read ‘a novel role of urinary’.    

Introduction

Line 37: Please define what ‘low environmental Cd exposure’ means in terms of daily exposure to Cd in micrograms per day.

Line 38: After ‘low dose Cd’ please provide the daily dose so the reader has better information about the actual exposure dose.    

Line 41: ‘with < 0.21 ug/L’ is unclear. Please clarify that this was the blood Cd concentration of an unexposed control population.

Line 43: Please remove ‘the’ before ‘U.S.’.

Line 48: It should read ‘the U.S.’.

Line 54: Please remove ‘)’ before ‘≥300’.

Line 62: It should read ‘examine if gender differences are related to Cd-induced hypertension’.

Materials and Methods

Line 67: After ‘diabetes type 2’ a reference or website should be provided so the reader is provided with the information about the ‘archived data’ that were analyzed in the present study.

Line 77: ‘metal work’ is a bit vague. Can this be more precisely worded so it is clearer as to what is meant?

Line 88: Information about the anticoagulant that was used to generate the blood plasma must be provided.

Line 90: It should read ‘for the concentration’.

Line 91:  The analytical methods that were used to determine the concentrations of Cd and b2M in urine should be provided here.

Line 95: It is unclear why only information is provided for the Cd determination in urine, but not how the concentration of b2M was measured? In addition, more details pertaining to the ‘atomic absorption spectrometry’ apparatus that was used must be provided (it should not read spectrophotometry). Presumably, a graphite furnace AAS instrument was used. If so the GFAAS model and manufacturer, wavelength, temperature program that was used for the analysis of the urine samples must be provided. It should be specified how many ul were injected into he graphite furnace and if  a matrix modifier was used for analysis? If so detailed must be provided. In addition, details about the graphite cuvettes that were used (standard?) should be provided (model number and manufacturer). All of these details are critical details to reproduce the study, which is the standard of any research study that involves measurements of any kind.

Line 97: Details about the ‘Multielement standards’ must be provided. Which metals does this solution contain and at what concentrations? What was the acid  composition of this standard.

Line 118: Please remove the blank before ‘normalization’.

Line 124: Please explain to the reader briefly what a ‘mediation model’ is and what it is typically used for.

Line 211: It should read ‘in relation to Cd’.

Line 261: It should read ‘Mediators of Cd-induced’.

Line 273: It should read ‘effect of Cd by’.

Line 277: It should read ‘mediating effect of Cd by’.

Line 285: It should read ‘were mediated through’.

Line 331: It should read ‘appeared to be mediated by’.

Line 351: Please replace ‘more susceptible’ by a more quantitative statement (e.g. 5-fold more susceptible…).

Line 379: ‘236,263,413’ is entirely unclear. Please correct accordingly.

Line 385: It should read ‘Ca and Cd’.

Line 386: It should read ‘Ca can be expected’.

Lien 387: It should read ‘effect of Ca has been’.    

Author Response

Reviewer 1 Round 2

Comments and Suggestions

The author has adequately addressed my previous suggestions to improve the overall clarity of the manuscript. There are, however, two remaining major deficiencies. The first is the failure to provide the reader with a reference or website which contains the data from the original study that were re-analyzed in the present study. While some cryptic information is provided as to how It is unclear to the reader what methods was analyzed to analyze blood and urine for b2-microglobulin. Secondly, any details that pertain to the analytical instruments that were employed to analyze the blood and urine samples are missing. These details must have been reported in the original study and should be briefly summarized in the present study, either in form of a Table and or electronic supplementary information. It is critical to provide the reader with detailed information about how the analytical data that are analyzed in the present study were generated as otherwise the analysis of the data could be inherently flawed. After addressing these major issues, the author should address some additional minor deficiencies before the manuscript can be accepted for publications. 

RESPONSES:

  • Thank you for your feedback and additional comments and suggestion for improvement of a paper.
  • Great effort and care have been taken to address, where possible, the issues the reviewer has raised.
  • Changes to the text are in red.

Detailed comments

Abstract

Line 9: It is suggested to provide the corresponding threshold Cd dose/day that links human Cd exposure with hypertension (e.g. ≥50 μg Cd per day).

Line 17+18: The abbreviation ‘POR’ as well as the 4th number after ‘POR’ must be explained.

Line 22: It would be helpful if the phrase ‘mediation model analysis’ could be briefly explained.

Line 23: It should read ‘a novel role of urinary’.   

RESPONSE:  Necessary corrections have been undertaken.

Introduction

Line 37: Please define what ‘low environmental Cd exposure’ means in terms of daily exposure to Cd in micrograms per day.

Line 38: After ‘low dose Cd’ please provide the daily dose so the reader has better information about the actual exposure dose.   

Line 41: ‘with < 0.21 ug/L’ is unclear. Please clarify that this was the blood Cd concentration of an unexposed control population.

Line 43: Please remove ‘the’ before ‘U.S.’.

Line 48: It should read ‘the U.S.’.

Line 54: Please remove ‘)’ before ‘≥300’.

Line 62: It should read ‘examine if gender differences are related to Cd-induced hypertension’.

RESPONSE:  Necessary corrections have been undertaken.

Materials and Methods

Line 67: After ‘diabetes type 2’ a reference or website should be provided so the reader is provided with the information about the ‘archived data’ that were analyzed in the present study.

Line 77: ‘metal work’ is a bit vague. Can this be more precisely worded so it is clearer as to what is meant?

Line 88: Information about the anticoagulant that was used to generate the blood plasma must be provided.

Line 90: It should read ‘for the concentration’.

Line 91:  The analytical methods that were used to determine the concentrations of Cd and b2M in urine should be provided here.

Line 95: It is unclear why only information is provided for the Cd determination in urine, but not how the concentration of b2M was measured? In addition, more details pertaining to the ‘atomic absorption spectrometry’ apparatus that was used must be provided (it should not read spectrophotometry). Presumably, a graphite furnace AAS instrument was used. If so the GFAAS model and manufacturer, wavelength, temperature program that was used for the analysis of the urine samples must be provided. It should be specified how many ul were injected into he graphite furnace and if  a matrix modifier was used for analysis? If so detailed must be provided. In addition, details about the graphite cuvettes that were used (standard?) should be provided (model number and manufacturer). All of these details are critical details to reproduce the study, which is the standard of any research study that involves measurements of any kind.

Line 97: Details about the ‘Multielement standards’ must be provided. Which metals does this solution contain and at what concentrations? What was the acid composition of this standard.

Line 118: Please remove the blank before ‘normalization’.

Line 124: Please explain to the reader briefly what a ‘mediation model’ is and what it is typically used for.

Line 211: It should read ‘in relation to Cd’.

RESPONSE:

  • Most issues have been addressed with exception for the following specific comments.

Line 67: After ‘diabetes type 2’ a reference or website should be provided so the reader is provided with the information about the ‘archived data’ that were analyzed in the present study.

Response: Data collected for 1500 subjects have not yet been deposited to any public database. They have been reported in ref. 32, 42-45.  These references have now been indicated in the text (line75) in Section 2.1. Recruitment of participants.

Line 95: It is unclear why only information is provided for the Cd determination in urine, but not how the concentration of b2M was measured.

Response:  All assays including urinary β2M, serum and urinary creatinine have been provided (lines 103-106). Information on an anticoagulant used and adjustment of urine pH to prevent β2M degradation in acidic conditions have been added (lines 97-100)

Results and Discussion

Line 261: It should read ‘Mediators of Cd-induced’.

Line 273: It should read ‘effect of Cd by’.

Line 277: It should read ‘mediating effect of Cd by’.

Line 285: It should read ‘were mediated through’.

Line 331: It should read ‘appeared to be mediated by’.

Line 351: Please replace ‘more susceptible’ by a more quantitative statement (e.g. 5-fold more susceptible…).

Line 379: ‘236,263,413’ is entirely unclear. Please correct accordingly.

Line 385: It should read ‘Ca and Cd’.

Line 386: It should read ‘Ca can be expected’.

Lien 387: It should read ‘effect of Ca has been’.   

RESPONSE:  Necessary corrections have been undertaken.

Round 3

Reviewer 1 Report

Comments and Suggestions for Authors

The author has addressed some of my minor suggestions to improve the overall clarity of the manuscript, but failed to address three major deficiencies.

  1. The author did not provide – as specifically requested - a reference or website which contains the data from the original study that were re-analyzed in the present study. The authors states in Line 72: ‘The study analyzed archived s=data from 689 Thai Nationals…’, but fails to provide any details to the previous study where these data have been published.
  2. The revised manuscript still contains only cryptic information about the analytical methods that were used to analyze blood and urine for creatinine and b2-microglobulin. Please see my detailed comment to the lack of information that is provided in Line 105 below.
  3. The previously requested instrumental analytical details about the analysis of urine and plasma sampled for Cd has not been provided: Below please find my request from my response to revised manuscript 2 below:

‘In addition, more details pertaining to the ‘atomic absorption spectrometry’ apparatus that was used must be provided (it should not read spectrophotometry). Presumably, a graphite furnace AAS instrument was used. If so the GFAAS model and manufacturer, wavelength, temperature program that was used for the analysis of the urine samples must be provided. It should be specified how many ul were injected into he graphite furnace and if  a matrix modifier was used for analysis? If so detailed must be provided. In addition, details about the graphite cuvettes that were used (standard?) should be provided (model number and manufacturer). All of these details are critical details to reproduce the study, which is the standard of any research study that involves measurements of any kind.’.

The instrumental analytical details must have been reported in the original study that the ‘archived data’ are from and have to be briefly summarized in the present study (either in form of a Table and or electronic supplementary information) as these details are critical to assess the validity of the reported concentrations. The statement in Line 110-111 reads ‘The analytical accuracy of the metal detection was checked by an external quality assessment every three years’. Based on the aforementioned deficiencies I reject the paper based on severe analytical deficiencies.

Detailed comments

Abstract

Line 18: IT is unclear what number ’83.4’ and ‘4.87’ refer. Please provide an explanation.    

Introduction     

Line 43: It should read ‘However, there is evidence that links an increased risk of CKD with exposure to a Cd dose between 10-15 ug/day [15-18].

Materials and Methods

Line 73: After ‘diabetes type 2’ a reference or website should be provided so the reader is provided with the information about where the ‘archived data’ that were analyzed in the present study were previously published. If the data were previously published in references 32, and/or 42-45 it should be explicitly stated.  

Line 99:  Please provide details as to how the samples were stored until they were analyzed (at what temperature?). The analytical methods that were used to determine the concentrations of Cd and b2M in urine should be provided here. It should read ‘Plasma was prepared from whole blood using…’.

Line 101: It should read ‘assayed for the concentration’.

Line 102: It should read ‘as detailed in previous reports’.

Line 104: Just stating that ‘Jaffe’s reaction’ was used (to measure the creatinine concentration) is entirely insufficient for a research paper. An adequate reference must be provided to the procedure that was used to analyze urine and plasma samples. In addition, the reader must be provided with a brief description how urine or blood samples were actually analyzed and the analytical instrument that was used to measure these concentrations (including the manufacturer and instrument model) must be specified.

Line 105: An adequate reference must be provided to the ‘immunoagglutination method’ based on which urine was analyzed for b2M. Just stating ‘LX test, Eiken 2MGII; Eiken and Shionogi Co, Tokyo, Japan’ is insufficient as it does not allow the reader to reproduce the study, which is the internationally accepted standard for any research. In addition, the author must provide a brief description about how the actual samples were analyzed and the analytical instrument that was used to measure these concentrations (including the manufacturer and instrument model) must be specified.

Line 107: Please remove ‘the’ and provide the GFAAS model that was used to analyze the samples, as this is critical information to reproduce the study if one wanted to.

Line 97: Details about the ‘Multielement standards’ must be provided. Which metals does this solution contain and at what concentrations? What was the acid composition of this standard.

Line 399: It is suggested to remove ‘236,263,413’.

Author Response

Response to comments on round-2 revisions from reviewer 1

Comments and Suggestions

The author has addressed some of my minor suggestions to improve the overall clarity of the manuscript, but failed to address three major deficiencies.

  1. The author did not provide – as specifically requested - a reference or website which contains the data from the original study that were re-analyzed in the present study. The authors states in Line 72: ‘The study analyzed archived s=data from 689 Thai Nationals…’, but fails to provide any details to the previous study where these data have been published.

RESPONSE:  References to which the data were published have been placed where the reviewer suggested (lines 71-74). To demonstrate data reporting transparency, references to selected subjects have been added (lines 74-81).  These new clarification statements are quoted below together with cited references.

This study analyzed archived data from 689 Thai Nationals (251 males, 438 females) who did not have diabetes type 2 [33, 34, 42-45], and were selected from a large population-based cohort of 1500 persons, conducted in compliance with the principles outlined in the Declaration of Helsinki. Of a total 689 subjects, 308 were residents of a Cd contaminated area of the Mae Sot District, Tak Province [33,34], and 381 persons were selected from the following three low-exposure localities; a non-contaminated part of the Mae Sot District (n = 63), Bangkok (n = 248) and Nakhon-Si-Thammarat Province (n = 70) [42-45]. For the Bangkok group, those aged 16 years or older were selected [42]. The health status was ascertained by physician’s examination reports and routine blood and urinary chemistry profiles. For the Nakhon-Si-Thammarat group [45], and the two Mae Sot groups, those had resided at their current addresses for 30 years or longer were selected [33,34,44].

[33]. Boonprasert, K.; Vesey, D.A.; Gobe, G.C.; Ruenweerayut, R.; Johnson, D.W.; Na-Bangchang, K.; Satarug, S. Is renal tubular cadmium toxicity clinically relevant? Clin. Kidney J. 2018, 11, 681–687.

[34]. Satarug, S.; Boonprasert, K.; Gobe, G.C.; Ruenweerayut, R.; Johnson, D.W.; Na-Bangchang, K.; Vesey, D.A. Chronic exposure to cadmium is associated with a marked reduction in glomerular filtration rate. Clin. Kidney J. 2018, 12, 468-475.

[42] Satarug, S.; Swaddiwudhipong, W.; Ruangyuttikarn, W.; Nishijo, M.; Ruiz, P. Modeling cadmium exposures in low- and high-exposure areas in Thailand. Environ. Health Perspect. 2013, 121, 531–536.

[43] Teeyakasem, W.; Nishijo, M.; Honda, R.; Satarug, S.; Swaddiwudhipong, W.; Ruangyuttikarn, W. Monitoring of cadmium toxicity in a Thai population with high-level environmental exposure. Toxicol. Lett. 2007, 169, 185-195.

[44] Satarug, S.; Đorđević, A.B.; Yimthiang, S.; Vesey, D.A.; Gobe, G.C. The NOAEL Equivalent of Environmental Cadmium Exposure Associated with GFR Reduction and Chronic Kidney Disease. Toxics 2022, 10, 614.

[45] Yimthiang, S.; Pouyfung, P.; Khamphaya, T.; Kuraeiad, S.; Wongrith, P.; Vesey, D.A.; Gobe, G.C.; Satarug, S. Effects of Environmental Exposure to Cadmium and Lead on the Risks of Diabetes and Kidney Dysfunction. Int. J. Environ. Res. Public Health 2022, 19, 2259.

  1. The revised manuscript still contains only cryptic information about the analytical methods that were used to analyze blood and urine for creatinine and b2-microglobulin. Please see my detailed comment to the lack of information that is provided in Line 105 below.

RESPONSE:  It is acknowledged that analytical methodologies to determine concentrations of creatinine and β2-microglobulin in serum and urine samples are provided in sufficient details for reproducibility purposes.  Accordingly, references to analytical methods used to measure β2M, creatinine concentrations have been provided (lines 102-105), quoted below.  

“Urinary and plasma creatinine concentrations were measured by the colorimetric method, based on the alkaline picrate Jaffe’s reaction, as described by Bonness and Taussky [49] and Apple et al. [50]. Urinary β2M assay was based on the latex immunoagglutination method, as described by Bernard et al. [51], using the LX test, Eiken 2MGII (Eiken and Shionogi Co., Tokyo, Japan).”

[49] Bonness, R.W.; Taussky, M.H. On the colorimetric determination of creatinine by the Jaffe reaction. J. Biol. Chem. 1945, 158, 581-591.

[50] Apple, F.; Bandt, C.; Prosch, A.; Erlandson, G.; Holmstrom, V.; Scholen, J., Googins M. Creatinine clearance: enzymatic vs Jaffé determinations of creatinine in plasma and urine. Clin. Chem. 1986, 32, 388-390.

[51] Bernard, A.M.; Vyskocil, A.; Lauwerys, R.R. Determination of beta 2-microglobulin in human urine and serum by latex immunoassay. Clin. Chem. 1981, 27, 832-837.

References 49, 50 and 51 provide assay principles and methodological details necessary to reproduction of the assays for creatinine and β2M concentrations.  In current practice, various commercial kits for β2M and creatinine assays are available along with methodological instructions provides by the manufactures.  Also, an automation system is available in clinical chemistry laboratories.  Hence, it serves no useful purposes to provide methodological details, which can be found from manufacturers’ guidance.

  1. The previously requested instrumental analytical details about the analysis of urine and plasma sampled for Cd has not been provided: Below please find my request from my response to revised manuscript 2 below:

‘In addition, more details pertaining to the ‘atomic absorption spectrometry’ apparatus that was used must be provided (it should not read spectrophotometry). Presumably, a graphite furnace AAS instrument was used. If so the GFAAS model and manufacturer, wavelength, temperature program that was used for the analysis of the urine samples must be provided. It should be specified how many ul were injected into he graphite furnace and if a matrix modifier was used for analysis? If so detailed must be provided. In addition, details about the graphite cuvettes that were used (standard?) should be provided (model number and manufacturer). All of these details are critical details to reproduce the study, which is the standard of any research study that involves measurements of any kind.’.

The instrumental analytical details must have been reported in the original study that the ‘archived data’ are from and have to be briefly summarized in the present study (either in form of a Table and or electronic supplementary information) as these details are critical to assess the validity of the reported concentrations. The statement in Line 110-111 reads ‘The analytical accuracy of the metal detection was checked by an external quality assessment every three years’. Based on the aforementioned deficiencies I reject the paper based on severe analytical deficiencies.

RESPONSE

  • It is acknowledged that instrumental information and analytical methodology to determine urinary Cd concentrations are provided in sufficient details for reproducibility purposes. Accordingly, a reference to analytical method and instrumentation used to measure urinary Cd concentrations has been provided together with multielement standard and application of reference urine samples for accuracy and precision ascertainment (lines 107-117), as quoted below.

“Urinary Cd concentration, expressed as µg/L was determined by Graphite Furnace Atomic Absorption Spectrometry (GBC System 5000, PAL2000 autosampler) [GBC Scientific Equipment, Hampshire, IL, USA], as described by Trzcinka-Ochocka et al. [51]. All test tubes, bottles, and pipettes used in metal analysis were acid-washed and rinsed thoroughly with deionized water. Multielement standards containing nine elements in dilute nitric acid, including As, Be, Cd, Cr (VI), Hg, Ni, Pb, Se, and Tl at 100 mg/L concentration for each element (Merck KGaA, Darmstadt, Germany) were used to calibrate the instrument. For quality control, the analytical accuracy and precision of Cd analysis were evaluated using reference urine metal controls containing Cd in three concentration ranges (Lyphocheck, Bio-Rad, Hercules, CA, USA). To maintain analytical accuracy, external quality assessments for Cd detection were conducted every three years.

The limit of detection (LOD) for urinary Cd was 0.1 µg/L. This LOD figure was 3 times the standard deviation of a repeated measurement of blank samples. When a urine sample contained Cd below its LOD, the Cd concentration assigned was the LOD value divided by the square root of 2 [53].

[52] Trzcinka-Ochocka, M.; Brodzka, R.; Janasik, B. Useful and Fast Method for Blood Lead and Cadmium Determination Using ICP-MS and GF-AAS; Validation Parameters. J. Clin. Lab. Anal. 2016, 30, 130-139.

  • Instrumental setting details such as emission wavelength are not provided because they serve no useful purposed and they can be found from manufacturers’ instruction.
  • However, it is utmost important to note that an interlaboratory comparability of reported urinary Cd concentrations is a critical issue, especially in meaningful epidemiological research studies. In the present study, such comparability was ascertained by using reference metal urine samples. Analytical accuracy and precision assurance were described in the original manuscript, and were maintained in both revision rounds. Reported Cd excretion rates found to be associated with an increased risk of hypertension were in ranges with meta-analysis by Verzelloni et al. [14]. This is a testimony of a standard Cd analysis practice of the present study.

[14] Verzelloni, P.; Giuliano, V.; Wise, L.A.; Urbano, T.; Baraldi, C.; Vinceti, M.; Filippini, T. Cadmium exposure and risk of hypertension: a systematic review and dose-response meta-analysis. Environ. Res. 2024, 120014.

Detailed comments

Abstract

Line 18: IT is unclear what number ’83.4’ and ‘4.87’ refer. Please provide an explanation.    

RESPONSE: The number 83.4 and 4.87 refer to the upper bound of 95% confidence interval values.  To reflect that referred numbers represent 95% confidences intervals, the pertaining statement has been changed to read as below (lines 16-18).

“Respective prevalence odds ratio (POR) and 95% confidence interval (CI) values for β2-microglobulinuria and hypertension were 10.7 (1.36−83.4), p = 0.024 and 2.79 (1.60−4.87) p <0.001, comparing the top quartile of ECd with the bottom quartile.

Introduction

Line 43: It should read ‘However, there is evidence that links an increased risk of CKD with exposure to a Cd dose between 10-15 ug/day [15-18].

RESPONSE:  A suggested replacement has been undertaken. 

Materials and Methods

Line 73: After ‘diabetes type 2’ a reference or website should be provided so the reader is provided with the information about where the ‘archived data’ that were analyzed in the present study were previously published. If the data were previously published in references 32, and/or 42-45 it should be explicitly stated.  

RESPONSE:  Please see responses to deficiency 1 above.

Line 99:  Please provide details as to how the samples were stored until they were analyzed (at what temperature?). The analytical methods that were used to determine the concentrations of Cd and b2M in urine should be provided here. It should read ‘Plasma was prepared from whole blood using…’.

RESPONSE:  It was stated in the original submission and was maintained in both rounds of revision that samples were stored at -80 ºC for later analysis.  

Line 101: It should read ‘assayed for the concentration’.

RESPONSE:  A typo error has been corrected.

Line 102: It should read ‘as detailed in previous reports’.

RESPONSE:  A typo error has been corrected.

Line 104: Just stating that ‘Jaffe’s reaction’ was used (to measure the creatinine concentration) is entirely insufficient for a research paper. An adequate reference must be provided to the procedure that was used to analyze urine and plasma samples. In addition, the reader must be provided with a brief description how urine or blood samples were actually analyzed and the analytical instrument that was used to measure these concentrations (including the manufacturer and instrument model) must be specified.

RESPONSE:  Please see responses to deficiency 2 above.

Line 105: An adequate reference must be provided to the ‘immunoagglutination method’ based on which urine was analyzed for b2M. Just stating ‘LX test, Eiken 2MGII; Eiken and Shionogi Co, Tokyo, Japan’ is insufficient as it does not allow the reader to reproduce the study, which is the internationally accepted standard for any research. In addition, the author must provide a brief description about how the actual samples were analyzed and the analytical instrument that was used to measure these concentrations (including the manufacturer and instrument model) must be specified.

RESPONSE:  Please see responses to deficiency 2 above.

Line 107: Please remove ‘the’ and provide the GFAAS model that was used to analyze the samples, as this is critical information to reproduce the study if one wanted to.

RESPONSE:  Please see responses to deficiency 3 above.

Line 97: Details about the ‘Multielement standards’ must be provided. Which metals does this solution contain and at what concentrations? What was the acid composition of this standard.

RESPONSE:  The elemental composition of multielement standard solution and their concentrations have been provided (lines 111-112).

Line 399: It is suggested to remove ‘236,263,413’.

 RESPONSE:  The referred numbers have been deleted.
